# *SERPINE1* rs6092 Variant Is Related to Plasma Coagulation Proteins in Patients with Severe COVID-19 from a Tertiary Care Hospital

**DOI:** 10.3390/biology11040595

**Published:** 2022-04-14

**Authors:** Ingrid Fricke-Galindo, Ivette Buendia-Roldan, Leslie Chavez-Galan, Gloria Pérez-Rubio, Rafael de Jesús Hernández-Zenteno, Espiridión Ramos-Martinez, Armando Zazueta-Márquez, Felipe Reyes-Melendres, Aimé Alarcón-Dionet, Javier Guzmán-Vargas, Omar Andrés Bravo-Gutiérrez, Teresa Quintero-Puerta, Ilse Adriana Gutiérrez-Pérez, Alejandro Ortega-Martínez, Enrique Ambrocio-Ortiz, Karol J. Nava-Quiroz, José Luis Bañuelos-Flores, María Esther Jaime-Capetillo, Mayra Mejía, Jorge Rojas-Serrano, Ramcés Falfán-Valencia

**Affiliations:** 1HLA Laboratory, Instituto Nacional de Enfermedades Respiratorias Ismael Cosío Villegas, Mexico City 14080, Mexico; ifricke@iner.gob.mx (I.F.-G.); glofos@yahoo.com.mx (G.P.-R.); armando.zazueta@iner.gob.mx (A.Z.-M.); luisf.reyes@iner.gob.mx (F.R.-M.); guzman.vargas.javier@gmail.com (J.G.-V.); a.bravo.gtz@gmail.com (O.A.B.-G.); teresa.quintero.fb@uas.edu.mx (T.Q.-P.); ilse.gutierrez@iner.gob.mx (I.A.G.-P.); alex_om_scv@outlook.com (A.O.-M.); e_ambrocio@iner.gob.mx (E.A.-O.); knava@iner.gob.mx (K.J.N.-Q.); 2Translational Research Laboratory on Aging and Pulmonary Fibrosis, Instituto Nacional de Enfermedades Respiratorias Ismael Cosio Villegas, Mexico City 14080, Mexico; ivettebu@yahoo.com.mx (I.B.-R.); adionet4@gmail.com (A.A.-D.); 3Laboratory of Integrative Immunology, Instituto Nacional de Enfermedades Respiratorias Ismael Cosio Villegas, Mexico City 14080, Mexico; lchavezgalan@gmail.com; 4COPD Clinic, Instituto Nacional de Enfermedades Respiratorias Ismael Cosío Villegas, Mexico City 14080, Mexico; rafherzen@yahoo.com.mx; 5Unidad de Investigación en Medicina Experimental, Facultad de Medicina, Universidad Nacional Autónoma de México, Mexico City 06720, Mexico; espiri77mx@yahoo.com; 6Clinical Laboratory Service, Instituto Nacional de Enfermedades Respiratorias Ismael Cosío Villegas, Mexico City 14080, Mexico; joseluisbf@yahoo.com.mx (J.L.B.-F.); tetejaime_2001@yahoo.com (M.E.J.-C.); 7Interstitial Pulmonary Diseases and Rheumatology Unit, Instituto Nacional de Enfermedades Respiratorias Ismael Cosio Villegas, Mexico City 06720, Mexico; medithmejia1965@gmail.com (M.M.); jrojas@iner.gob.mx (J.R.-S.)

**Keywords:** COVID-19, coagulation, genetics, *F5*, *SERPINE1*, PSGL-1, P-selectin

## Abstract

**Simple Summary:**

Severe forms of coronavirus disease 2019 (COVID-19) are related to an alteration in the coagulation process determined by genetic factors. Variability on *F5* and *SERPINE1* genes has been previously reported as a risk factor for thrombosis associated with different diseases. In this study, we aimed to evaluate if two relevant variants in the *F5* and *SERPINE1* genes were related to the requirement of invasive mechanical ventilation in hospitalized patients with COVID-19 and/or the levels of coagulation-related proteins. We observed that patients presenting the studied variants in *F5* and *SERPINE1* exhibit different levels of coagulation-related proteins. Moreover, the levels of these proteins were higher among patients requiring invasive mechanical ventilation when compared to hospitalized patients with no ventilation support. This study contributes to the genetics insight regarding COVID-19 severity and the inter-individual susceptibility for a severe form of the infectious disease.

**Abstract:**

An impaired coagulation process has been described in patients with severe or critical coronavirus disease (COVID-19). Nevertheless, the implication of coagulation-related genes has not been explored. We aimed to evaluate the impact of *F5* rs6025 and *SERPINE1* rs6092 on invasive mechanical ventilation (IMV) requirement and the levels of coagulation proteins among patients with severe COVID-19. Four-hundred fifty-five patients with severe COVID-19 were genotyped using TaqMan assays. Coagulation-related proteins (P-Selectin, D-dimer, P-selectin glycoprotein ligand-1, tissue plasminogen activator [tPA], plasminogen activator inhibitor-1, and Factor IX) were assessed by cytometric bead arrays in one- and two-time determinations. Accordingly, *SERPINE1* rs6092, P-Selectin (GG 385 pg/mL vs. AG+AA 632 pg/mL, *p* = 0.0037), and tPA (GG 1858 pg/mL vs. AG+AA 2546 pg/mL, *p* = 0.0284) levels were different. Patients carrying the CT *F5*-rs6025 genotype exhibited lower levels of factor IX (CC 17,136 pg/mL vs. CT 10,247 pg/mL, *p* = 0.0355). Coagulation proteins were also different among IMV patients than non-IMV. PSGL-1 levels were significantly increased in the late stage of COVID-19 (>10 days). The frequencies of *F5* rs6025 and *SERPINE1* rs6092 variants were not different among IMV and non-IMV. The *SERPINE1* rs6092 variant is related to the impaired coagulation process in patients with COVID-19 severe.

## 1. Introduction

Since December 2019, the coronavirus disease 2019 (COVID-19) has affected more than 180 million people worldwide, and, unfortunately, it has led to more than 5 million deaths [1]. A broad clinical spectrum of COVID-19 has been reported, including mild, moderate, severe, and critical forms of the disease [2,3]. According to an extensive Chinese cohort report, most patients presented a mild disease (81%), while 14% and 5% progressed to severe and critical COVID-19 [4]. A recent meta-analysis including 281,461 individuals reported 22.9% severe illness, while the mortality was 5.6% [5].

The main initial symptoms include fever, cough, dyspnea, malaise, fatigue, and sputum or secretion [6]. In adults, the severe COVID-19 clinical manifestations start approximately seven days after the onset of the symptoms [7]. They can include dyspnea, respiratory rates of 30 ≥ breaths per minute, blood oxygen saturation ≤ 93%, a ratio of the partial pressure of arterial oxygen to the fraction of inspired oxygen (PaO_2_/FiO_2_) < 300, or infiltrates in more than 50% of the lung field [4].

Several risk factors for developing a critical or severe COVID-19 have been identified. Age is considered the most critical condition for the severity and mortality of COVID-19. In addition, men tend to present more complications than women as well as individuals with chronic diseases such as cardiovascular disease, diabetes mellitus, immunosuppression, and obesity [8]. The most severe forms of COVID-19 require admission to the intensive care unit and, probably, invasive mechanical ventilation (IMV) due to lung injury and acute respiratory distress syndrome [9].

Different studies have reported an altered coagulation activity in patients with severe COVID-19, regardless of anticoagulant prophylaxis therapy [10]. A meta-analysis reported a prevalence of venous thromboembolism (VTE), deep venous thrombosis (DVT), and pulmonary embolism (PE) of 28.4%, 25.6%, and 26.4%, respectively, among critical patients with COVID-19 [11]. Moreover, another meta-analysis reported that patients with COVID-19 who developed VTE presented a higher mortality rate than those who did not present VTE [12].

The mechanisms of the thrombosis complications in patients with severe COVID-19 remain unknown. However, evidence from different studies suggests that it is a consequence of the cytokine release syndrome with activation of leukocytes and platelets, resulting in the up-regulation of tissue factors, activation of the coagulation process, thrombin generation, and fibrin formation. In addition, coagulation is also affected by the imbalance of plasminogen activator-1 (PAI-1), tissue factor pathway inhibitor, and activated protein C that promotes fibrin generation and the inhibition of fibrinolysis [11].

Numerous conditions predispose individuals to venous thrombosis, such as aging, prolonged immobilization, surgery, fractures, oral contraceptives, hormone replacement therapy, pregnancy, puerperium, cancer, and antiphospholipid syndrome [13]. In addition, variants in genes related to the coagulation process have been identified as genetic risk factors for venous thrombosis. For instance, the genes encoding factor V (*F5*) and the serine proteinase inhibitor of tissue plasminogen activator (*SERPINE1*) have been associated with thrombosis risk in Caucasian populations [14] and with thrombosis related to other diseases such as cancer [15], ischemic stroke [16], and pregnancy loss [17].

Factor V accelerates the conversion of prothrombin to thrombin, which is the main enzyme of the coagulation pathway, and begins the fibrin clot and the aggregation of platelets [18]. Meanwhile, the product of *SERPINE1*, PAI-1, inhibits fibrinolysis by inhibiting the plasminogen activator. PAI-1 is stored in the platelets and is produced by endothelial cells, megakaryocytes, smooth muscle cells, fibroblasts, monocytes, adipocytes, hepatocytes, and other cell types [19].

Identifying genetic biomarkers could be a clue in the early diagnosis and treatment of severe COVID-19. We aimed to evaluate *F5* rs6025 and *SERPINE1* rs6092’s impact on IMV among patients with severe COVID-19. In addition, the relation of these variants with levels of coagulation-related proteins was assessed in a subgroup of patients.

Herein, we reported a difference in the levels of coagulation proteins in patients with severe COVID-19 according to *F5* rs6025 and *SERPINE1* rs6092 genetic variants. Moreover, we observed the relevance of coagulation-related proteins in the clinical course of COVID-19. This study contributes to the knowledge of the disease and identifies individuals requiring specific attention to apply anticoagulation strategies.

## 2. Materials and Methods

Four hundred fifty-five patients hospitalized in the Instituto Nacional de Enfermedades Respiratorias Ismael Cosio Villegas (Mexico City, Mexico) and diagnosed with COVID-19 were included in this study. Only patients with a positive SARS-CoV-2 RT-PCR test and ≥18 years old were consecutively enrolled and signed informed consent. The study protocol was approved by the local Research Ethics Committee (C53-20) and complied with the Helsinki Declaration statements.

A severe case of COVID-19 was identified in all the included patients considering they presented dyspnea, a respiratory rate of ≥30 breaths per minute, blood oxygen saturation ≤ 90%, and PaO_2_/FiO_2_ ≤ 300. Therefore, patients were stratified into two critical groups for the association study according to whether they required IMV or not (non-IMV). All patients were treated with enoxaparin in thromboprophylaxis doses or therapeutic anticoagulation when admitted to the hospital, based on standardized recommendations and not on the degree of critical illness [3,20]. Thromboprophylaxis was indicated for all hospitalized patients that warranted it. The risk of venous thromboembolism was stratified according to the scale modified Caprini risk assessment model and the risk of bleeding using the IMPROVE bleeding risk model [20].

### 2.1. Genetic Analysis

Genomic DNA was isolated by standard techniques from peripheral blood samples and collected in tubes with EDTA as an anticoagulant. DNA samples were verified for purity and integrity and stored at 4 °C until use. The *F5* rs6025 and *SERPINE1* rs6092 variants were genotyped using the TaqMan^®^ SNP Genotyping Assays (C__11975250_10 and C___2620926_10, respectively), according to the supplier instructions, in a 7300 Real-Time PCR System (Applied Biosystems, Carlsbad, CA, USA).

### 2.2. Coagulation Proteins Determination

Proteins related to coagulation were determined in a subgroup of the total patients included in the study. The selection criteria included the genotypes for *F5* rs6025 and *SERPINE1* rs6092 (to assure genetic heterogeneity) and the time of the sample collection after patients were admitted. Thus, the proteins were assessed in the plasmas of 131 patients collected within their first 0–9 days of hospital admission (mean 3.3 ± 2 days). Additionally, in 35 of these patients, a second sample was taken 10–24 days after admission (12.6 ± 1.3 days) (Appendix A). Plasma was separated from blood samples collected in EDTA tubes by centrifugation at 4500 rpm for 5 min and stored at −80 °C until assayed.

The quantification of P-selectin, D-dimer, PSGL-1 (P-selectin glycoprotein ligand-1), tPA (tissue plasminogen activator), PAI-1, and factor IX was performed using the LEGENDplexTM Human Thrombosis Panel (7-plex) (BioLegend^®^, San Diego, CA, USA). Multiparametric flow cytometry was performed using a FACS Aria II flow cytometer (Becton Dickinson, San Jose, CA, USA).

### 2.3. Statistical Analysis

Categorical variables are presented as frequencies and continuous variables as mean ± standard deviation (SD), or median and interquartile range [IQR] if the values were not normally distributed. The Kolmogorov-Smirnov test was performed to assess the normality.

The allele and genotype differences among the critical groups were assessed using a Fisher’s exact. In addition, a binary logistic regression was carried out to adjust for severity risk variables found for COVID-19 (leucocyte and platelets). Both association tests were performed using PLINK v1.07 [21].

The levels of coagulation proteins were assessed for correlation with leucocyte, lymphocyte, platelets, age, hospitalization days, smoking index, body mass index (BMI), and PaO_2_/FiO_2_, using a Spearman rank correlation test. In addition, the Mann-Whitney U test was performed to evaluate the differences in the proteins’ levels according to the *F5* rs6025 and *SERPINE1* rs6092 genotypes and the critical groups (IMV and Non-IMV). A delta value was calculated for each protein level in the two-time determination groups (first determination—second determination). Wilcoxon determined the differences in this delta value among genotype and critical groups signed-rank test. The statistical significance was set at a *p* < 0.05, and the tests were performed using RStudio v. 1.3.1073 [22].

## 3. Results

The demographic and clinical data of the patients are shown in Table 1. Males were predominant among the included patients, although no significant difference was observed. In addition, 82.8% of patients had a BMI > 25.0 kg/m^2^, from which 48% presented obesity (BMI ≥ 30.0 kg/m^2^) and 34.8% were overweight (BMI 25.0–29.9 kg/m^2^). Approximately a third of the patients were smokers and had diabetes and/or hypertension.

Leukocyte and platelets counts and the PaO_2_/FiO_2_ index were significantly different between the studied groups. Patients requiring IMV had a more extended stay in the hospital than non-IMV. The remaining available variables were not found to have statistically significant differences.

In addition, the differences in the coagulation proteins’ levels among IMV and non-IMV groups were evaluated (Table 2). Levels were higher in IMV patients than in non-IMV, except for P-Selectin, showing the most significant risk of coagulation abnormalities among COVID-19 patients requiring IMV.

### 3.1. Genetic Association Study with COVID-19 Severity

First, we evaluated the association of genetic variants with the COVID-19 severity considering the critical groups (IMV and non-IMV). Table 3 shows the genotype and allele frequencies of *F5* rs6025 and *SERPINE1* rs6092 variants calculated in the study. No differences were observed among the IMV, non-IMV groups (Table 3), or the logistic regression model when the adjustment for co-variables was performed (Appendix A).

### 3.2. Levels of PSGL-1 and tPA Were Different According to SERPINE1 rs6092 Genotypes

Coagulation proteins were determined in the plasma of 131 patients with severe COVID-19, and the association with *F5* rs6025 and *SERPINE1* rs6092 was assessed. For the *SERPINE1* rs6092 variant, P-Selectin and tPA levels were significantly different among GG carriers vs. AG+AA genotypes. In addition, lower levels of Factor IX were observed for the patients with the CT genotype of *F5* rs6025 compared to CC carriers (Figure 1, Table 4); however, further studies are required to confirm this finding since the CT group only included the two patients carrying the alternative allele of the *F5* rs6025. The levels of the remaining proteins were not different among the studied genotypes (Table 4, Appendix A).

The correlation studies of the proteins’ levels with clinical variables were performed for different subgroups and presented as correlation plots in Figure 2 and Figure 3. In the correlation plots, including the subgroup of one-time determination, we observed correlations within the levels of the coagulation proteins (Figure 2). The highest ρ value was observed for the correspondence of PAI-1 with tPA levels. In addition, the leukocyte count was negatively correlated with PaO_2_/FiO_2_ (ρ = −0.23, *p* = 0.038) (Figure 2a), which agrees with the findings in Table 1 (lower levels of leukocyte in non-IMV patients with higher PaO_2_/FiO_2_). Interestingly, a correlation between PSGL-1 and P-Selectin was observed (ρ = 0.43, *p* = 0.0299) when only patients with the GG *SERPINE1* genotype were included in the correlation study (Figure 2b). Meanwhile, the analysis comprising only patients with AG+AA genotypes showed a correlation of age with D-dimer levels (ρ = 0.32, *p* = 0.0033), as well as a negative correlation between the platelets count and hospitalization days (ρ = −0.35, *p* = 0.0282) (Figure 2c). In addition, we evaluated if the proteins’ levels were different among females and males, but there were no significant differences (Appendix A).

The proteins’ levels from the subgroup of two-times determination were heterogeneous. As a whole, the proteins’ levels tend to increase in the second determination after 10–24 days of the patients’ admission to the hospital, but this was not significantly different for most of the proteins except for PSGL-1 (Table 5). Nevertheless, this was not accomplished in all the studied subjects since, for some patients, the proteins’ levels were lower in the second determination than in the initial quantification. This finding was independent of the groups, and there was no difference in the proteins’ levels considering the critical groups (IMV and Non-IMV) and the *SERPINE1* rs6092 genotypes (Appendix A). Likewise, the delta values of the two determinations were not significantly different among the study groups (data not shown).

Apparently, in the late stage of COVID-19, the coagulation proteins’ levels are influenced by different factors. This can be observed in the correlation plots of the delta values (Figure 3a) and the second determination (Figure 3b), in which there are several weak and moderate correlations of proteins’ levels with clinical data. For instance, in both cases, BMI correlated with Factor IX levels, and the delta value of this protein was also correlated with age and lymphocyte count. In addition, platelets count correlated with PaO_2_/FiO_2_. Nevertheless, the sample size from this subgroup is different from the one-time determination, which could be a bias in the observed results. In addition, considering the time for the second assessment of coagulation proteins, the influence of the clinical interventions (pharmacological and non-pharmacological) could be expected.

## 4. Discussion

Early identification of patients with severe or critical COVID-19 at risk of complications could decrease the mortality of these patients and optimize health services. To the best of our knowledge, this is the first study reporting the association of *F5* rs6025 and *SERPINE1* rs6092 variants with the levels of coagulation proteins related to COVID-19 complications.

*F5* has been identified as one of the critical genes involved in coagulation during COVID-19 infection [23]. A systematic review described *F5*, *FGA*, *FGB*, and *FGG* interacting and influencing D-dimer and fibrinogen levels [24]. Likewise, in bioinformatic analysis, *F5* was identified as a gene shared among COVID-19 comorbidities (kidney disease, liver disease, diabetes, lung disease, and cardiovascular disease) [25]. However, *F5* variants had not been previously evaluated in patients with COVID-19. Our study did not find an association of the rs6025 variant with the IMV requirement in patients with severe COVID-19; however, we observed differences in Factor IX levels among patients with CC and CT *F5* genotypes. Unfortunately, we could not assess factor V levels, and although there is an inter-relation of the coagulation factors [26], the results should be taken with caution since the frequency of CT genotypes was extremely low. Factor IX levels have been previously related to thrombosis risk [27], and we also observed that it was different among IMV and non-IMV groups. This factor plays an essential role in blood coagulation, considering the bleeding tendency associated with congenital factor IX deficiency, and, after its activation, it is involved in thrombin generation in the vicinity of platelets [28].

*SERPINE1* encodes PAI-1, which inhibits fibrinolysis; thus, an excess of this protein is related to thrombophilia [29]. The rs6092 variant is located at position 43 (c.43G>A), leading to a change of alanine to threonine in the residue 15 (p.Ala15Thr) [30]. One study has suggested that the AA and AG genotypes are related to higher plasma PAI-1 levels than the GG genotype [31]. Although this was not significant, we also observed increased levels of PAI-1 in plasma of AG+AA genotypes compared to GG carriers. Meanwhile, tPA levels were different among patients considering the rs6092 genotype, and the levels of this protein strongly correlated (Figure 2, ρ > 0.90, *p* < 0.05) with PAI-1 levels. This suggests a relation between the *SERPINE1* rs6092 variant and PAI-1 levels, but this might not be observed due to other factors not evaluated in this study that could be affecting the plasma PAI-1. Increased plasma PAI-1 in diabetes, metabolic syndrome, insulin resistance, and obesity has been reported [32]. In addition, other genetic variants in *SERPINE1*, such as rs1799889, have been related to PAI-1 expression and the risk and protective effect of different diseases [32].

Plasma levels of P-Selectin were also significantly different according to rs6092 *SERPINE1*. The P-Selectin level has been related to coronary artery disease in patients with diabetes mellitus [33], and it has also been found to increase in ICU patients with COVID-19 [34]. It was proposed as a diagnostic and prognostic biomarker for COVID-19 [35]; however, we did not observe differences in the levels of proteins among IMV and non-IMV groups, but the selection criteria in both studies present relevant differences. A relation or interaction of *SERPINE1* and the P-Selectin gene (*SELP*) has not been described. Therefore, further studies are warranted to evaluate if *SERPINE1* rs6092 can influence P-Selectin levels or indirectly correlates with other coagulation proteins that we could not observe in our research. For instance, a moderate correlation of PAI-1 with P-Selectin levels has been observed in patients with rheumatoid arthritis [36].

The impairment of the coagulation process in COVID-19 has been widely described. We found significantly different levels of all the evaluated coagulation proteins (D-dimer, PSGL-1, tPA, PAI-1, and Factor IX), except for P-Selectin, among IMV and non-IMV groups. The increase of D-dimer levels has been mainly correlated with COVID-19 severity and in-hospital mortality [37]. PSGL-1 was not previously determined in patients with COVID-19, but it has been reported that this glycoprotein is incorporated into virion particles, inhibiting the virion attachment to target cells [38,39]. Moreover, this glycoprotein has been described as a critical regulatory molecule in the activation of platelets, leukocytes, and vascular endothelial cells; and the expression of tissue factors on leukocytes, the serum levels of inflammatory factors, fibrinogen deposition, as well as other immunologic and coagulation disturbances, are alleviated when the glycoprotein is blockaded with a PSGL-1 antibody in endotoxemic mice [40]. Furthermore, the expression of this protein in microvesicles has been associated with unprovoked venous thromboembolism [41]. In our study, the level of PSGL-1 was found to increase in IMV patients and was the only coagulation protein that showed a significant increase in the second determination of coagulation proteins in plasma levels, which could indicate the relevance of this glycoprotein in the late stage of COVID-19. It also presented a moderate correlation with D-dimer, PAI-1, and Factor IX (ρ > 0.50 Figure 3b), contributing to the critical status of patients with COVID-19.

Additionally, tPA and PAI-1 are involved in the impaired fibrinolytic activities in patients with acute respiratory distress syndrome due to COVID-19 [42] and worse respiratory status [43,44]. The administration of tPA has been proposed to improve oxygenation in patients with severe COVID-19 [43,45]. Factor IX has not been previously determined in the plasma of patients with COVID-19. Nevertheless, an investigation measured total levels of von Willebrand factor (vWF) and vWF binding to platelet glycoprotein complex (GpIb-IX-V) in patients with COVID-19 from an intensive care unit, and both forms of the vWF were markedly increased in these patients [46].

Although the coagulation proteins’ levels increased in the second determination in most cases, the clinical and pharmacological intervention during the hospitalization stay could explain the lack of significant variability observed for these parameters measured within 10–15 days of the first determination. Nevertheless, the studied biomarkers for mortality prediction in the late stage of COVID-19 (>14 days) are focused on blood cells (i.e., leukocytes, lymphocytes, and platelets) that are involved in the coagulation process [37]. Activated platelets recruit more platelets and secrete proinflammatory cytokines and proangiogenic factors, which promote leukocyte activation and extravasation. Macrophages generate plasmin to degrade fibrin into D-dimer. Activated monocytes produce inflammatory cytokines and chemokines, which stimulate neutrophils, lymphocytes, platelets, vascular endothelial cells, and monocytes to express tissue factors and phosphatidylserine and trigger coagulation [37,47]. Moreover, abnormal platelet count has previously been related to IMV requirement and different complications in patients with community-acquired pneumonia admitted to intensive care units [48,49] and COVID-19, in which increased platelet consumption has been suggested [48].

Regarding blood cells, we found significant differences in leukocytes and platelets counts among IMV and non-IMV groups, but not for lymphocytes. Higher leukocytes levels can be due to a coinfection, as observed in patients with COVID-19 and a positive culture for bacteria and fungus, associated with high mortality risk in COVID-19 patients [49,50]. It has been reported that the enhanced inflammatory state, thrombi formation, and platelet consumption can lead to thrombocytopenia, while the cytokine storm causes thrombocytosis [37].

This study presents some limitations. We lacked information regarding complete treatment before and during the hospitalization stay since the treatment guidelines have changed according to the knowledge of the disease and each patient’s requirements during the pandemic. Moreover, other clinical parameters and studies for diagnosing thrombosis would have contributed significantly to the present findings. Likewise, we could not determine the coagulation proteins in a control group. Finally, the low frequency of *F5* rs6025 requires further studies with larger sample sizes to draw solid conclusions as to its relevance in the levels of coagulation-related proteins and the severity of COVID-19.

## 5. Conclusions

We observed differences in the coagulation proteins’ levels according to *F5* and *SERPINE1* genotypes and among IMV and non-IMV groups, indicating an increased risk of complications due to an impaired coagulation activity in patients with severe COVID-19. Identifying patients with severe or critical COVID-19 at risk of complications could improve the treatment and outcome of these patients and optimize health services.

## Figures and Tables

**Figure 1 biology-11-00595-f001:**
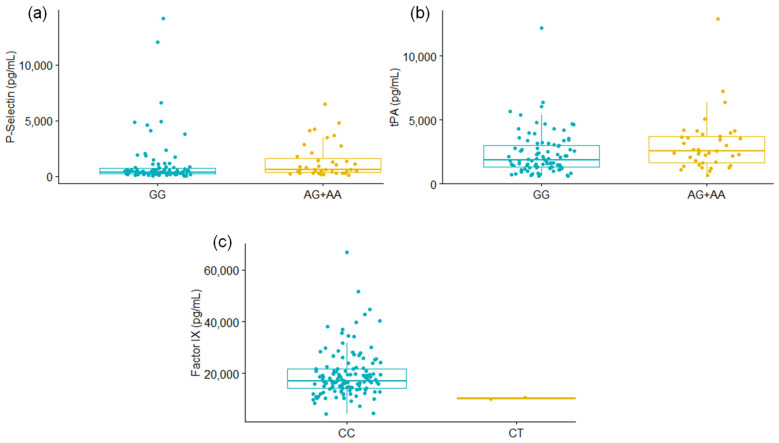
Differences in P-Selectin (**a**) and tPA (**b**) levels according to the *SERPINE1* rs6092 genotype; and of Factor IX levels (**c**) when the genotype of *F5* rs6025 was considered. Mann-Whitney U Test. Only results with significant values are shown.

**Figure 2 biology-11-00595-f002:**
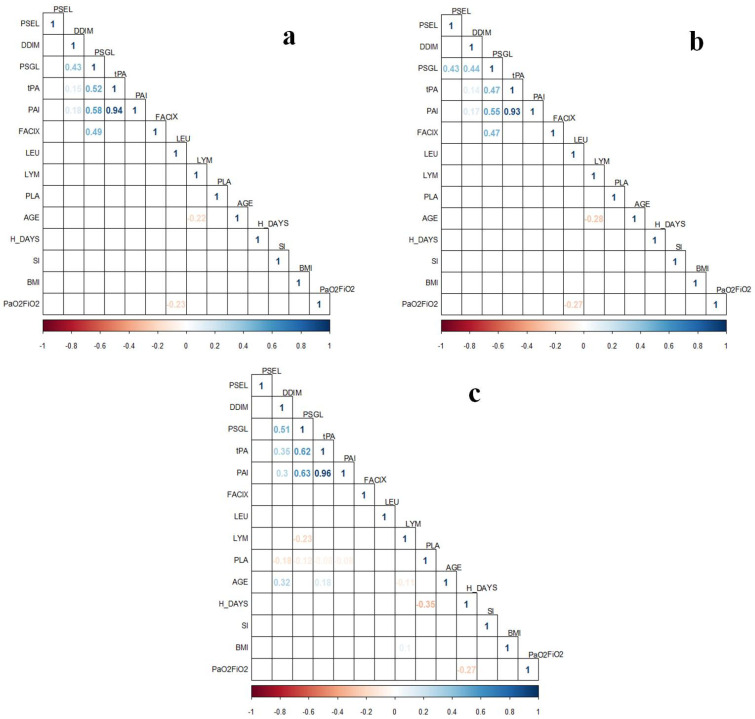
Correlation plots of coagulation proteins’ levels from one-time determination subgroup (*n* = 131) with clinical variables in patients with COVID-19. (**a**) Analysis including all samples of the one-time determination subgroup (*n* = 131); (**b**) Analysis only including patients carrying the *SERPINE1* rs6092 GG genotype (*n* = 92); (**c**) Analysis only including patients carrying the *SERPINE1* rs6092 AG+AA genotypes (*n* = 39). Correlations were assessed with Spearman Correlation Test. Only statistically significant ρ values (*p* < 0.05) are shown in the plots. BMI, body mass index; FACIX, Factor IX; DDIM, D-dimer; H_DAYS, hospitalization days; LEU, leukocytes; LYM, lymphocytes; PAI, plasminogen activator inhibitor-1; PLA, platelets; PSGL, P-selectin glycoprotein ligand-1; PSEL, P-selectin; SI, smoking index; tPA, tissue plasminogen activator.

**Figure 3 biology-11-00595-f003:**
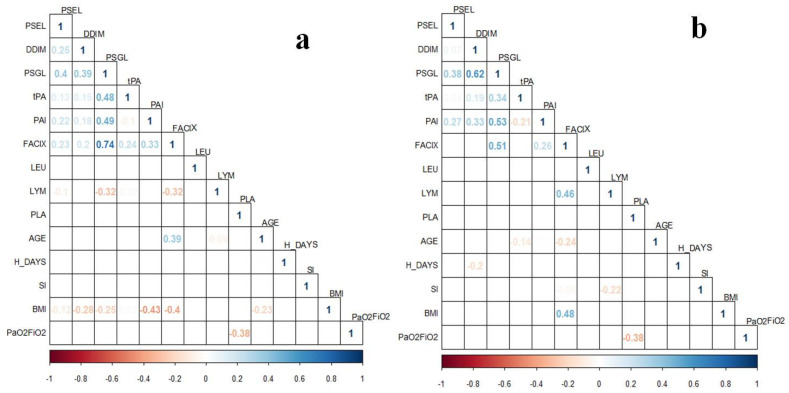
Correlation plots of coagulation proteins’ levels from the two-time determination subgroup (*n* = 35) with clinical variables in patients with COVID-19. (**a**) Coagulation proteins correspond to the delta values assessed (Time 1–Time 2); (**b**) Coagulation proteins’ levels from the second determination (Time 2). Correlations were assessed with Spearman Correlation Test. Only ρ values statistically significant (*p* < 0.05) are shown in the plots. BMI, body mass index; FACIX, Factor IX; DDIM, D-dimer; H_DAYS, hospitalization days; LEU, leukocytes; LYM, lymphocytes; PAI, plasminogen activator inhibitor-1; PLA, platelets; PSGL, P-selectin glycoprotein ligand-1; PSEL, P-selectin; SI, smoking index; tPA, tissue plasminogen activator.

**Table 1 biology-11-00595-t001:** Demographic and clinical data of patients with COVID-19 included in the study.

	All (*n* = 455)	IMV (*n* = 331)	Non-IMV (*n* = 124)	*p*-Value *
Age, yr	58 [49–67]	58 [49–68]	56 [48–65.3]	0.188
Gender, F/M *n* (%)	144 (31.6)/311 (68.4)	98 (29.6)/233 (70.4)	46 (37.1)/78 (62.9)	0.141
BMI, kg/m^2^	29.7 [26.1–33.1]	29.9 [26.5–33.1]	28.9 [25.8–33.1]	0.466
Smoking, *n* (%)	117 (25.7)	83 (25.1)	34 (27.4)	0.697
Hospitalization Days	18 [12–30]	21 [16–35]	11 [8.3–15]	**<0.001**
Symptoms Onset, days	9 [7–12]	9 [7–13]	9 [6–11]	0.236
Diabetes, *n* (%)	138 (30.3)	106 (32.0)	32 (25.8)	0.200
Hypertension, *n* (%)	152 (33.4)	110 (33.2)	42 (33.9)	1.000
Leukocyte, [10^9^/L]	9.6 [7.2–12.9]	10.6 [7.7–13.7]	7.9 [5.9–9.9]	**<0.001**
Lymphocyte, [10^9^/L]	0.7 [0.5–1.2]	0.7 [0.5–1.2]	0.8 [0.6–1.2]	0.095
Platelets, [10^9^/L]	280 [214–356]	273 [210.8–343.2]	307 [226.8–393]	**0.018**
IMV Days	10 [0–21]	18 [11–28.5]	NA	NA
PaO_2_/FiO_2_	166 [114–214]	149 [103–193.2]	213.5 [165.8–248.0]	**<0.001**

Continuous data are presented as median [interquartile range] and frequencies of categorical data in absolute numbers and percentages. * *p*-value from Mann-Whitney U Test or Exact Fisher’s Test. Statistically significant values are presented in bold style. BMI, body mass index; F, female; IMV, invasive mechanical ventilation; M, male; yr, year.

**Table 2 biology-11-00595-t002:** Differences in coagulation proteins’ levels according to IMV and Non-IMV groups (*n* = 131).

Protein (pg/mL)	IMV (*n* = 94)	Non-IMV (*n* = 37)	*p*-Value *
P-Selectin	415 [221–1004]	533 [291–837]	0.3225
D-dimer	586 [313–1140]	355 [261–534]	**0.0156**
PSGL-1	6016 [5083–7175]	5058 [4612–6414]	**0.0210**
tPA	2309 [1502–3620]	1517 [1274–2395]	**0.0044**
PAI-1	81,326 [73,927–88,980]	76,344 [72,747–80,737]	**0.0122**
Factor IX	17,956 [14,651–22,189]	15,821 [12,710–18,935]	**0.0365**

Proteins’ levels are presented as median [interquartile range]. * Mann-Whitney U Test. Statistically significant values are presented in bold style. IMV, invasive mechanical ventilation; PAI-1, plasminogen activator inhibitor-1; PSGL-1, P-selectin glycoprotein ligand-1; tPA, tissue plasminogen activator.

**Table 3 biology-11-00595-t003:** Association study of genotype and allele frequencies of *F5* rs6025 and *SERPINE1* rs6092 among critical groups.

Variants	All (*n* = 455)	IMV (*n* = 331)	Non-IMV (*n* = 124)	*p*-Value *
*F5* rs6025				
CC	453 (0.996)	329 (0.994)	124 (1.000)	1.000
CT	2 (0.004)	2 (0.006)	0	
C	908 (0.998)	660 (0.997)	248 (1.000)	1.000
T	2 (0.002)	2 (0.003)	0	
*SERPINE1* rs6092				
GG	400 (0.879)	290 (0.876)	110 (0.887)	0.632
AG	54 (0.119)	40 (0.121)	14 (0.113)	
AA	1 (0.002)	1 (0.003)	0	
G	854 (0.938)	620 (0.937)	234 (0.943)	0.876
A	56 (0.061)	42 (0.063)	14 (0.057)	

* Fisher’s Exact Test. IMV, invasive mechanical ventilation.

**Table 4 biology-11-00595-t004:** Differences in the proteins’ levels according to *SERPINE1* rs6092 and *F5* rs6025 genotypes (*n* = 131).

**Protein (pg/mL)**	***SERPINE1* rs6092**		***p*-Value ***
	**GG (*n* = 92)**	**AG+AA (*n* = 39)**	
P-Selectin	385 [215–720]	632.3 [358–1617]	**0.0037**
D-dimer	508 [275–1020]	468 [296–1022]	0.8582
PSGL-1	5822 [4899–7034]	6008 [4995–7171]	0.3128
tPA	1858 [1310–3017]	2546 [1674.3–3689]	**0.0284**
PAI-1	78,027 [72,987–87,230]	80,737 [75,598–86,422]	0.1647
Factor IX	16,890 [14,287–21,699]	18,363 [13,732–20,751]	0.7152
	***F5* rs6025**		
	**CC (*n* = 129)**	**CT (*n* = 2)**	
P-Selectin	447 [243–951]	1392 [904–1880]	0.4250
D-dimer	485 [277–1001]	1498 [1168–1828]	0.1406
PSGL-1	5944 [4931–7045]	6334 [5812–6856]	0.6796
tPA	2112 [1406–3227]	2447 [1840–3055]	0.9701
PAI-1	78,763 [73,462–87,210]	81,797 [79,485–84,109]	0.7073
Factor IX	17,136 [14,323–21,662]	10,247 [10,108–10,386]	**0.0355**

Proteins’ levels are presented as median [interquartile range]; * Mann-Whitney U Test. Statistically significant values are presented in bold style. IMV, invasive mechanical ventilation; PAI-1, plasminogen activator inhibitor-1; PSGL-1, P-selectin glycoprotein ligand-1; tPA, tissue plasminogen activator.

**Table 5 biology-11-00595-t005:** Two-time determination of coagulation proteins in patients with severe COVID-19 (*n* = 35).

**Protein (pg/mL)**	**Time 1 (0–9 Days)**	**Time 2 (10–24 Days)**	***p*-Value ***	**Δ (Time 1–Time 2)**
P-Selectin	416 [226–781]	482 [164–880]	0.3547	−72 [−603–223]
D-Dimer	660 [341–1654]	922 [486–1915]	0.1032	−183 [−1012–171]
PSGL-1	5944 [4940–6884]	6830 [5307–8810]	**0.0059**	−1371 [−3237–270]
tPA	2093 [1457–3487]	2295 [1471–3581]	0.1819	−399 [−2014–881]
PAI-1	80,462 [74,035–87,832]	78,763 [70,988–93,465]	0.7370	−2718 [−1800–8437]
Factor IX	17,621 [14,833–23,071]	19,413 [11,519–24,126]	0.8506	−397 [−4729–5452]

Proteins’ levels are presented as median [interquartile range]; * Wilcoxon signed-rank test. Statistically significant values are presented in bold style. PAI-1, plasminogen activator inhibitor-1; PSGL-1, P-selectin glycoprotein ligand-1; tPA, tissue plasminogen activator.

## Data Availability

Accession ClinVar database: SCV001750078 and SCV001750079.

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
