# Peer review of "SERPINE1* rs6092 Variant Is Related to Plasma Coagulation Proteins in Patients with Severe COVID-19 from a Tertiary Care Hospital"

_biology, 2022, doi:10.3390/biology11040595_

Round 1
Reviewer 1 Report
The decision is on you. The authors have responded to my comments.
Author Response
Thank you for your comments.
The manuscript was reviewed and corrected since the initial evaluation and also improved for grammar and general redaction.
Reviewer 2 Report
The present study is very interesting, well done and supported by clear and obvious results.
The present study addresses a problem of great interest regarding the identifications of risk factors for developing a critical or severe COVID-19 and the demonstration of an association of an altered coagulation activity in patients with severe COVID-19. The aim was to evaluate F5 rs6025 and SERPINE 1 rs6092 impact on IMV among patients with severe COVID-19. In addition, the relation of these variants with levels of coagulation-related proteins was assessed in a subgroup of patients. Results showed differences in the coagulation protein levels according the F5 and SERPINE1 genotypes, as well as among IMV and Non-IMV groups, indicating an increased risk of complications due to an impaired coagulation activity in patients with severe COVID-19. This study contributes to the knowledge of the disease and the identification of patients with severe or critical COVID-19 at risk of complications could improve the treatment and outcome of these patients and optimize health services.
Author Response
Thank you for your comments.
The manuscript was improved for grammar and general redaction.
This manuscript is a resubmission of an earlier submission. The following is a list of the peer review reports and author responses from that submission.
Round 1
Reviewer 1 Report
In this manuscript, Fricke-Galindo et al aimed to prove that SERPINE1 Rs6092 variant is related to plasma coagulation proteins in patients with severe COVID-19 from a Tertiary Care Hospital. The authors observed differences in the coagulation proteins levels according to the genes encoding factor V (F5) and the serine proteinase inhibitor of tissue plasminogen activator (SERPINE1) genotypes and also between invasive mechanical ventilation (IMV) and non-IMV groups. From these results, they suggested that an increased risk of complications can occur in an impaired coagulation activity in patients with severe COVID-19. I found this report is interesting but it would fit better in a clinical journal. Thus, I do not recommend this manuscript to be published at the Cells.
There are strange errors that make the manuscript so difficult to read. The authors should improve their writing before resubmission.
The quality of the manuscript can be significantly improved if the results are presented as Figures.
Reviewer 2 Report
The authors evaluate the effect of two genetic variants, rs6025 and rs6092, and several coagulation proteins among patients requiring mechanical ventilation compared with those not requiring mechanical ventilation in severe COVID-19.
The major concern regarding this manuscript is the clinical relevance and novel information in the manuscript. The correlation with coagulation factors is well described in covid19, and the current study is not powered to evaluate the association between the studied genetic variants given the low incidence in the study population.
Heterozygosity for rs6025 is described in 0.04% of the study cohort. Is this the expected level of inheritance in a south American/Mexican population? With relatively small numbers in the study, it is not possible to draw conclusions regarding the role of the FVL mutation in severe COVID19 and, in particular, in mechanical ventilation.
The risk alleles for the variants are not defined and the manuscript does not demonstrate which alleles are associated with having the genetic variants. Also, it is not clear why single alleles are included for both rs6025 and for rs6092. It would seem there should only be 3 combinations (ie. XX, XY and YY). The table should also include the total numbers of the genetic variants seen in the population.
The clinical relevance of the parameters is not defined? Is there reference ranges suggested? It is difficult to understand the meaning of the variables without a range.
Do the authors have access to a non-severe COVID19 control group?
Other
In the methods section, can the authors confirm how many of the variables were required to meet the definition of severe COVID1-19?
In the table with the genotypes and allele frequencies, rs6025 is 0.004 in all, 0.006 in IMV and 0 in non-IMV. Can the authors please clarify these findings?